

# Estimating daily surface NO$_2$ concentrations from satellite data - A case study over Hong Kong using land use regression models

Jasdeep S Anand[1] and Paul S Monks[1]

[1]Atmospheric Chemistry Group, Department of Chemistry, University of Leicester, University Road, Leicester, LE1 7RH, UK

*Correspondence to:* J. S. Anand (jsa13@le.ac.uk)

**Abstract.** Land Use Regression (LUR) models have been used in epidemiology to determine the fine-scale spatial variation in air pollutants such as nitrogen dioxide (NO$_2$) in cities and larger regions. However, they are often limited in their temporal resolution, which may potentially be rectified by employing the synoptic coverage provided by satellite measurements. In this work a mixed effects LUR model is developed to model daily surface NO$_2$ concentrations over the Hong Kong SAR during 2005-2015. In-situ measurements from the Hong Kong Air Quality Monitoring Network, along with tropospheric vertical column density (VCD) data from the OMI, GOME-2A and SCIAMACHY satellite instruments were combined with fine-scale land use parameters to provide the spatiotemporal information necessary to predict daily surface concentrations. Cross-validation with the in-situ data shows that the mixed effect LUR model using OMI data has a high predictive power (adj. R$^2$ = 0.84), especially when compared with surface concentrations derived using the MACC-II reanalysis model dataset (adj. R$^2$ = 0.11). Time series analysis shows no statistically significant trend in NO$_2$ concentrations during 2005-2015, despite a reported decline in NO$_x$ emissions. This study demonstrates the utility in combining satellite data with LUR models to derive daily maps of ambient surface NO$_2$ for use in exposure studies.

## 1 Introduction

It has been shown (WHO, 2013) that ambient exposure to outdoor nitrogen dioxide (NO$_2$) has long-term health impacts stemming from cardiovascular and respiratory illnesses. In rapidly urbanising countries such as China the cost of poor air quality is especially high (e.g. Chen et al., 2012; Gu et al., 2012). In particular, the Hong Kong Special Administrative Region (SAR) has seen significant economic growth in recent decades, which has resulted in the emergence of photochemical smog events caused by increased nitrogen oxide (NO$_x$) emissions. These effects have been further exacerbated by transported emissions and pollution from the nearby Pearl River Delta (PRD, Xue et al., 2014). It has previously been estimated that air quality improvement from the annual average to the lowest pollutant levels of better visibility days, comparable to the World Health Organization (WHO) air quality guidelines, would lead to 1335 fewer deaths a year over this region, with a saving of over US$240 million in both direct costs and productivity losses (Hedley et al., 2008).

Reliable exposure assessment requires constructing accurate maps of average pollutant concentrations. However, concentration data is often sourced from sparse in-situ measurements which are typically from regulatory monitoring networks. Mapping pollutant exposure therefore requires the spatial interpolation of these measurements over a fine scale, taking into account





known emission sources and sinks to estimate the true pollutant distribution. A possible technique to achieve this interpolation is Land Use Regression (LUR, Hoek et al., 2008), in which concentrations measured by in-situ stations are correlated with predictor variables such as traffic or population density using a Geographic Information System (GIS). A multivariate linear regression model is constructed based on significant covariates, which can then be used to estimate the pollutant concentration elsewhere.

LUR models are considered to be advantageous, as unlike dispersion modelling they do not require detailed information about atmospheric conditions as input data. As they are based on linear regression, LUR models are computationally inexpensive to run compared to dispersion modelling. Previously, LUR models have been used to model species such as $NO_x$ and particulate matter over spatial scales ranging from cities to countries (e.g. Beelen et al., 2013; Eeftens et al., 2012; Chen et al., 2010; Meng et al., 2015). However, most LUR models are limited by their temporal resolution, and are typically used to determine seasonal or annual concentrations. Methods to improve the temporal resolution of LUR models often involve rescaling temporally coarser models based on trends observed in regulatory monitoring data.

In addition to in-situ networks, $NO_2$ can also be measured from space by satellite instruments (Monks and Beirle, 2011). Satellite datasets have some advantages over in-situ networks, in that their long service life and revisit time can provide long-term monitoring of major emission sources and ambient atmospheric conditions, allowing for synoptic coverage of both spatial and temporal variation over urban areas. However, these instruments are only capable of measuring tropospheric vertical column densities (VCDs), and so cannot be readily compared with in-situ concentrations without accurately modelling the $NO_2$ vertical profile to separate the above-ground contribution (e.g. Bechle et al., 2013). Also, because of their coarse spatial resolution, satellites are not capable of resolving fine-scale urban variation. For instance, modelled $NO_2$ VCDs at the same spatial footprint as the Ozone Monitoring Instrument (OMI, Levelt et al., 2006) over North American megacities were found to have a 20-30% negative bias when compared to fine-scale models (Kim et al., 2016).

Data from satellites have previously been used in $NO_2$ LUR models over large geographic regions. For instance, average surface concentrations derived from tropospheric $NO_2$ VCDs measured by OMI have been used as predictor variables to estimate annual $NO_2$ concentrations over the United States (Novotny et al., 2011), Western Europe (Vienneau et al., 2013), and Australia (Knibbs et al., 2014). OMI tropospheric VCDs have also successfully been used directly without deriving a surface concentration to model the annual $NO_2$ concentration over The Netherlands (Hoek et al., 2015). In all cases the inclusion of OMI data as a predictor variable resulted in good agreement with in-situ measurements, and improved predictive performance when compared with equivalent LUR models which did not include OMI data.

The aforementioned examples can only provide time-averaged concentrations- and so may be sensitive to daily variations in $NO_2$ caused by changes in local meteorology or emission sources. Daily satellite measurements may contain useful information about both of these effects, and so could be applied to address this issue. Lee and Koutrakis (2014) used a mixed effects model to address this issue. In this LUR model, the OMI tropospheric VCD was included with both a fixed and random effects. Fixed effects representing parameters temperature and wind speed were also included, along with land use terms such as population density and developed area. The LUR model was found to have high predictive capability ($R^2 = 0.79$) when used to estimate daily $NO_2$ concentrations over the New England region of the USA.



A similar mixed effects approach could potentially be used to predict NO$_2$ concentrations over China. Because of limited data availability there have been few exposure assessment studies of Chinese air quality. A LUR model would allow for daily high-resolution maps to be developed for such studies. The objective of this work is to therefore create and validate a LUR model for forecasting surface NO$_2$ concentrations over Hong Kong, and to assess its utility.

## 2 Method

For this work surface NO$_2$ concentrations were measured and forecasted over the Hong Kong SAR between 2005-2015. This time period was chosen as a compromise between ensuring adequate representation of seasonal cycles and the availability and quality of the satellite data (see below).

### 2.1 In-situ data

The LUR models used in this work were both calibrated and validated by surface NO$_2$ concentrations measured by in-situ stations from the Hong Kong Air Quality Network (HK-AQN). these stations are maintained by the Hong Kong Environmental Protection Department (HKEPD, 2007). Between 2005-2015 11 monitoring stations measuring ambient pollutant concentrations were in operation (see Figure 1). These stations provide hourly measurements of CO, SO$_2$, O$_3$, NO$_x$, NO$_2$, and particulate matter. NO$_2$ concentrations are measured through a combination of chemiluminescence and Differential Optical Absorption Spectroscopy (DOAS, Platt and Stutz, 2006). These stations are placed on buildings, away from traffic junctions, and so are thought to be representative of ambient conditions. Of these stations, 10 are located in developed regions while one (Tap Mun) is located in the Sai Kung Country Park, and so can be considered a rural background station. Throughout the study period, the HKEPD have reported that the precision and accuracy of the NO$_2$ measurements have been within the $\pm 20\%$ control limit.

### 2.2 Satellite data

Between 2005-2015 there were three satellite instruments measuring tropospheric NO$_2$ VCDs. All retrieval algorithms based on these instruments derive these VCDs by first retrieving a total slant column density (SCD) from the measured visible (400-500 nm) reflectance spectrum using the DOAS technique. The stratospheric component of the total column is then separated, either by empirical estimation based on unpolluted regions (e.g. Richter and Burrows, 2002) or by model assimilation (e.g. Boersma et al., 2004). In addition to this, the column is also weighted by an air mass factor (Palmer et al., 2001) calculated from *a priori* information to account for biases resulting from scene-specific features (e.g. viewing geometry, scene albedo, NO$_2$ vertical profile).

Because of their varying ground pixel sizes, all satellite data products used in this work were reprojected onto a 0.01° grid. To avoid biases from cloud contamination, only ground pixels where the reported cloud fraction was <30% were used from all instruments. For scanning instruments, only pixels observed during foreward scans were used.





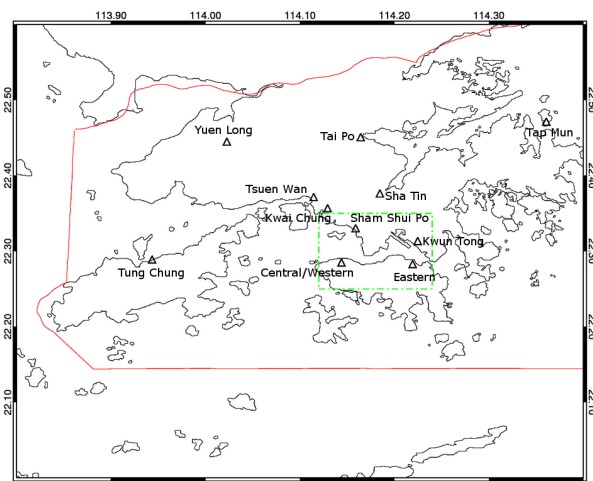

**Figure 1.** The in-situ $NO_2$ stations from the HK-AQN used in this work. The red line indicates the international boundary of the Hong Kong SAR, which this work focuses on. The green rectangle represents the Kowloon district and Hong Kong Island, from which the time series in Section 3.8 was derived.

### 2.2.1 Ozone Monitoring Instrument (OMI)

The Dutch-Finnish Ozone Monitoring Instrument (OMI, Levelt et al., 2006) has been in continuous operation since 2004. OMI daily global coverage, with a local equatorial overpass time of approximately 13:45. OMI observes a 2600 km swath binned to 60 across-track pixels. The nadir pixel size of OMI is $13\times24$ km$^2$, which allows for near-urban mapping of $NO_2$. However,

as OMI is a pushbroom spectrometer the pixel size increases considerably away from the nadir. To try and compensate for this effect, the ground pixels are weighted by their size and cloud fraction when gridded using the method detailed in Wenig et al. (2008).

Since 2007 OMI has also been affected by a partial blockage of its entrance aperture. This obstruction has resulted in the so-called "row anomaly", in which the measured radiances are systematically biased depending on the across-track viewing

angle, season, and latitude. At the time of this work this anomaly affects roughly half of the 60 across-track pixels, which are removed from the analysis.

For this work the OMI tropospheric VCDs were taken from the NASA Standard Product (OMNO2, v 3.0  OMNO2 Team, 2016). In this product the global stratospheric $NO_2$ field is estimated by interpolating over known unpolluted regions and then subtracted from the total column (Bucsela et al., 2013). Further information about the SCD fit and the AMF computation can

be found in Bucsela et al. (2013) and Marchenko et al. (2015).




### 2.2.2 Global Ozone Monitoring Experiment-2 (GOME-2A)

The Global Ozone Monitoring Experiment-2A (GOME-2A, Callies et al., 2000) has offered near-global coverage of tropospheric $NO_2$ since 2007. GOME-2A has a local equatorial overpass time of roughly 09:30, and observes a 1920 km swath using a scanning mirror. Because of this, the ground pixel size during the forward-scan remains $80\times40$ km$^2$ throughout the swath. From the launch of GOME-2B in 2013 the viewing configuration of GOME-2A was changed, such that the swath width was reduced to 960 km. While this has improved the spatial resolution to $40\times40$ km$^2$, daily global coverage is no longer possible from GOME-2A. Because of this change no data after 2012 is used in this work.

For this work the GOME-2A tropospheric VCDs were taken from the TEMIS TM4NO2A product (v 2.3 Boersma et al., 2004). In this product the total SCD is assimilated into the TM4 chemical transport model (CTM) to obtain the stratospheric column. Further information about the SCD fit and the AMF computation can be found in TEMIS (2010).

### 2.2.3 SCanning Imaging Absorption spectroMeter for Atmospheric CHartographY (SCIAMACHY)

The SCanning Imaging Absorption spectroMeter for Atmospheric CHartographY (SCIAMACHY, Bovensmann et al., 1999) was in operation between 2002-2012. SCIAMACHY used both limb and nadir viewing geometries to provide columnar and profile information. However, because of this unique design global coverage was only achieved every 6 days. SCIAMACHY had a local equatorial overpass time of 10:00. Like GOME-2A, SCIAMACHY employed a scanning mirror to image a 960 km swath, which allowed for a constant ground pixel size of $60\times30$ km$^2$.

For this work the SCIAMACHY tropospheric VCDs were also taken from the TEMIS TM4NO2A product (v 2.3 Boersma et al., 2004).

### 2.3 Mixed effects LUR model

The mixed effects LUR model considered in this work is similar to the one developed by Lee and Koutrakis (2014). The daily ambient $NO_2$ concentration at a location, $i$, on day, $j$, is assumed to be a linear function of the the gridded daily satellite tropospheric $NO_2$ VCD retrieved over the same location, $\Omega_{ij}$:

$$NO_{2,ij} = \alpha + u_j + (\beta_1 + v_j)\Omega_i + \sum_m \beta_m X_{ijm} + \epsilon_{ij} (u_j v_j) \sim N[(00), \Sigma] \tag{1}$$

This approach accounts for day-to-day variations in the $NO_2 \cdot \Omega$ relationship, while reducing the influence of days with insufficient in-situ or satellite data on the daily relationship.

In equation (1) $\alpha$ and $u_j$ are the fixed and random intercepts, respectively, while $\beta_1$ and $v_j$ are the fixed and random slopes of $\Omega_{ij}$, respectively. $\beta_m$ are the fixed slopes of additional predictor variables, $X_{ijm}$, at point, $i$, and day, $j$. The error term of the model is represented by, $\epsilon_{ij}(u_j v_j) \sim N(0, \sigma^2)$, while, $\Sigma$, represents the variance-covariance relationship for the day-specific random effects.



The fixed terms in equation (1) represent the spatial average of the $NO_2 \cdot \Omega$ relationship, while the random terms model the day-specific variations. The day-specific relationship may be the consequence of daily variations in the $NO_2$ vertical profile caused by changes in boundary layer height, emissions, or other influences. For this work the daily mean $NO_2$ concentration from each of the stations shown in Figure 1 was used as the dependent variable in equation (1). These concentrations were

log-transformed to ensure that the input dataset was normally distributed.

### 2.3.1 Spatial predictor variables

As in traditional LUR models, spatial predictor variables in this work are selected from a number of proxies describing the local meteorology and $NO_2$ emission sources and sinks. These are summarised in Table 1 and discussed herein. Variables describing sources and sinks at a given location were also buffered using several circle radii: 100, 200, 300, 400, 500, 600,

700, 800, 1000, 1200, 1500, 1800, 2000, 2500, 3000, 3500, 4000, 5000, 6000, 7000, 8000, and 10000 m. In all, this gave a total of 139 distinct variables to be presented to the model. Certain variables were also given fixed signs that $\beta_m$ must have. For instance, terms representing emission sources must have positive $\beta_m$ terms to represent the positive effect they have on the ambient $NO_2$ concentration, while variables such as vegetation cover and surface elevation would have a negative $\beta_m$.

At the time of this work no traffic density information for Hong Kong was available, so in order to estimate the possible con-

tribution from traffic emissions it was thought that the total road length within a buffer radius would be a viable substitute. Road lengths were calculated from the OpenStreetMap dataset (Haklay and Weber, 2008). The road lengths of primary, secondary, and tertiary roads were considered as separate variables to account for the average difference in traffic density experienced by these road types. The coastline from the OpenStreetMap dataset was also used to calculate the distance to the sea for a given point, in order to simulate the possible effect of ocean deposition and/or shipping emissions on the ambient $NO_2$ concentration.

Residential $NO_2$ emissions were thought to scale linearly with population density, which has been sourced from the World-Pop 2010 population density dataset (Stevens et al., 2015). The total population density within a buffer was calculated for a given point.

Urban area coverage was also assumed to be a good indicator of residential and industrial emissions. At the time of this work the highest resolution land cover dataset available over Hong Kong was the 0.5 km MODIS-based Global Land Cover

Climatology (Broxton et al., 2014). The total vegetation cover (i.e. land covered by any vegetation type) was also used to simulate the effect of dry deposition on the ambient $NO_2$ concentration. Both vegetation and urban cover were calculated as a percentage of the buffer area.

In addition to fixed spatial parameters Lee and Koutrakis (2014) also suggested using meteorological data in the model to further explain the spatiotemporal variation in the surface $NO_2$ field. For instance, surface temperature can be assumed to be

a proxy for solar radiation, and so the photochemical rate of dissociation of $NO_2$ into NO, while wind speed can be used as a proxy for the effect of advection on local concentrations. For this work the daily mean surface temperature, wind speed and wind direction sourced from the ERA-Interim reanalysis dataset (Dee et al., 2011) were used as predictor variables.





| Physical property | Variable | Type of variable | Data source (resolution) | Preferred sign | Reference |
|---|---|---|---|---|---|
| Vehicle emissions | Road length (Primary, secondary, tertiary) | Buffered (sum) | OpenStreetMap (N/A) | Positive | Haklay and Weber (2008) |
| Industrial emissions | Urban area coverage | Buffered (%) | MODIS-based Global Land Cover Climatology (0.5 km) | Positive | Broxton et al. (2014) |
| Residential emissions | Population density | Buffered (sum) | WorldPop dataset (1 km) | Positive | Stevens et al. (2015) |
| Dry deposition | Vegetation area coverage | Buffered (%) | MODIS-based Global Land Cover Climatology (0.5 km) | Negative | Broxton et al. (2014) |
| Ocean deposition | Distance from coast | Point | OpenStreetMap (N/A) | N/A | Haklay and Weber (2008) |
| Surface elevation | Surface elevation | Point | ASTER Global Digital Elevation Model V2 (30 m) | Negative | Tachikawa et al. (2011) |
| Surface temperature | Daily 2 m Temperature | Point | ERA-Interim reanalysis (0.125°) | Negative | Dee et al. (2011) |
| Wind advection | Daily wind direction and speed | Point | ERA-Interim reanalysis (0.125°) | N/A | Dee et al. (2011) |
| Location | Latitude and Longitude | Point | - | N/A | - |

**Table 1.** The predictor variables ($X_m$) considered for the LUR model (equation (1)) used in this work.

### 2.3.2 Predictor variable selection

To determine the optimal combination of predictor variables to be used in the LUR model, a stepwise regression approach similar to the one employed by Eeftens et al. (2012) was used. First, univariate regression was applied to all predictor variables. The predictor variable with the highest adjusted $R^2$ was included in the equation (1) as the first $X_m$. The remaining variables are then consecutively added to the model, and their effect on the model adjusted $R_2$ was noted. After all other variables are considered, the predictor variable that resulted in the largest increase in the adjusted $R_2$ was kept, provided that the following criteria are met: 1) The increase in the model adjusted $R^2$ was greater than 1%, 2) The sign of the predictor variable coefficient conformed to the sign shown in Table 1, 3) The signs of the other predictor variables already included in the model were not changed by the inclusion of the considered predictor variable.

Predictor variables were added to the model until the model adjusted $R^2$ no longer increased by >1%. The p-values of each predictor variable were then calculated, with statistically insignificant variables (i.e. $p > 0.05$) sequentially removed from the





model until all predictor variables became statistically significant. The multicollinearity of the remaining predictor variables was then assessed by calculating the variance inflation factor (VIF) for each one. Predictor variables where VIF >10 were sequentially removed from the model to determine their influence on the model predictive power.

The models developed for this work were also tested for influential observations by calculating the Cook's D for each surface $NO_2$ measurement. Observations where the Cook's D was >1 would be removed from the analysis and their effect on the model performance would have been assessed. However, in this work no such observations were detected over any of the stations involved.

### 2.3.3   Model variants

Daily forecasts of surface $NO_2$ will be affected by the diurnal and seasonal cycles that affect transport and production. Because
of their different revisit times, data from the satellite instruments have previously been combined to yield information about these cycles (e.g. Boersma et al., 2008; Hilboll et al., 2013). Therefore, it may be possible to enhance the model predictive power by using observations by multiple satellite instruments at the same time and location. Equation (1) can therefore be adapted to include random and fixed slopes and intercepts for each satellite instrument. For instance, a model combining SCIAMACHY and OMI data would be:

$$NO_{2,ij} = \alpha + u_{j,OMI} + u_{j,SCIA} + (\beta_{1,OMI} + v_{j,OMI})\,\Omega_{i,OMI} + (\beta_{1,SCIA} + v_{j,SCIA})\,\Omega_{i,SCIA} + \sum_m \beta_m X_{ijm} + \epsilon_{ij}\,(u_j v_j) \sim N\left[(00), \Sigma\right]$$

15                                                                                                                                        (2)

In this case the fixed and random slopes of $\Omega$ now represent the average and day-specific $NO_2 \cdot \Omega$ relationship as observed by each instrument, which may allow for the diurnal cycle to be better represented in the model.

Additionally, previous studies (e.g. Beelen et al., 2013) used separate LUR models to account for seasonality in surface $NO_2$ concentrations. While the use of daily satellite data should help to account for this effect, over short time scales the systematic
difference between seasons may not be immediately recognisable and may lead to a poor model fit.

For this work several models were developed to explore these concepts, which are summarised in Table 2. Model 1 is a reference against all other models are compared against, as the OMI dataset is the temporally longest with minimal issues from spatial sampling or cloud cover. Model 2 attempts to account for the seasonal cycle by training two LUR models looking at different months for all years: winter (November-April) and summer (May-October). Several LUR models are also trained to
investigate the predictive utility of each satellite instrument separately. In addition to this, several models based on equation (2) were assessed, trialling different combinations of satellite instruments in order to better account for diurnal variations in $NO_2$. Finally, a multiple linear regression model without using satellite data or mixed effects, while forcing temperature and wind speed as predictor variables, was also assessed as a reference to compare against the other models.

Other models based on those listed in Table 2 were also tested, but are not included in this work due to anomalous results.
A seasonal model similar to Model 2 was tested with GOME-2 and SCIAMACHY data, but in both cases the fixed satellite data slope was found to be statistically insignificant in the winter season. It is likely that this result was due to both instruments



| Model number | Time period | Satellite instrument(s) |
|---|---|---|
| 1 | 2005-2015 | OMI |
| 2 | 2005-2015: winter (Nov-Apr) summer (May-Oct) | OMI |
| 3 | 2005-2012 | SCIAMACHY |
| 4 | 2007-2013 | GOME-2A |
| 5 | 2007-2012 | GOME-2A + SCIAMACHY |
| 6 | 2007-2013 | GOME-2A + OMI |
| 7 | 2005-2012 | SCIAMACHY + OMI |
| 8 | 2007-2012 | GOME-2A + SCIAMACHY + OMI |
| 9 | 2005-2015 | N/A (Reference) |

**Table 2.** The LUR models considered in this work, showing the time period and satellite instruments used. Note that model 9 is a multiple linear regression model which does not include satellite data or random effects.

lacking an adequate number of winter measurements over Hong Kong because of their comparatively large ground pixel size and limited coverage.

## 3   Results and discussion

The properties of each of the models (predictor variables, adjusted $R^2$) discussed in Table 2 are summarised in Table 3. Comparisons between these models may be biased by the number of observations used to produce each model, owing to the difference in mission lifetimes and ground pixel sizes. Additionally, models combining OMI and SCIAMACHY data always failed to converge, regardless of the predictor variables included. This null result may be due to a lack of cloud-free days when both instruments were coincident over Hong Kong. Despite this, it is clear that models including satellite data have superior predictive performance as compared with the reference model.

Figure 2 shows the mean surface $NO_2$ concentration during 2005-2015 as predicted by Model 1, compared to the mean OMI tropospheric $NO_2$ VCD observed during the same period. The Model 1 output shows clear enhancements over known residential areas, with the densely populated districts of Kowloon, Wai Chung, and Kwai Chung showing concentrations $> 100$ $\mu gm^{-3}$. Additional enhancements are also visible over Hong Kong International Airport, and industrial parks such as Yantian. Conversely, unpopulated regions such as the Plower Cove and Sai Kung Country Parks show very low concentrations ($\sim 5$ $\mu gm^{-3}$). Outside of the Hong Kong SAR, significant enhancements are also found over Shenzhen and Bao'an, which likely reflect the high population density and manufacturing industries located there.





| Model number | Predictor variables (m) | N | Adjusted $R^2$ |
|---|---|---|---|
| 1 | Secondary (600) and Tertiary (300, 7000) road length, Longitude | 1610 | 0.828 |
| 2 (winter) | Primary road length (8000), Urban area (400), Longitude | 7493 | 0.804 |
| 2 (summer) | Secondary (500) and Tertiary (300, 3500, 7000) road length | 8667 | 0.797 |
| 3 | Secondary (1200) and Tertiary (300, 7000) road length, Longitude | 884 | 0.824 |
| 4 | Tertiary (300, 7000) road length, Population density (200), Longitude | 3777 | 0.854 |
| 5 | Primary (6000) and Tertiary (400) road length, Urban area (600), Longitude | 296 | 0.846 |
| 6 | Tertiary (300, 7000) road length, Population density (200), Longitude | 3369 | 0.860 |
| 8 | Primary (6000) and Tertiary (500) road length, Urban area (1000), Longitude | 216 | 0.863 |
| 9 | Tertiary (300, 500, 3500, 7000) road length, Longitude | 39159 | 0.419 |

**Table 3.** Description of the LUR models shown in Table 2, showing the predictor variables (including buffer radii where applicable) and adjusted $R^2$. Models combining OMI and SCIAMACHY data failed to converge regardless of predictor variable, so no viable dataset was produced for Model 7.

By contrast, the raw OMI data does not adequately resolve any of these features, showing only a single enhancement over Bao'an which declines radially with distance. This discrepancy is likely to be the consequence of poor spatial sampling and vertical mixing being dominated by emissions from mainland China. The difference in detail between these two datasets shows the potential utility in downscaling coarse satellite data with mixed effects LUR models to better resolve emission sources and spatial distribution.

## 3.1 Model intercomparison

Figure 3 shows the mean surface $NO_2$ concentration predicted by all models between 2007-2012, which was a time period common to all of them. Because of differences in instrument spatial resolution and ground coverage, only 38 days in this time period were found to have cloud-free measurements by all three satellite instruments. As a compromise, Figure 3 shows the mean of all data predicted by each model.





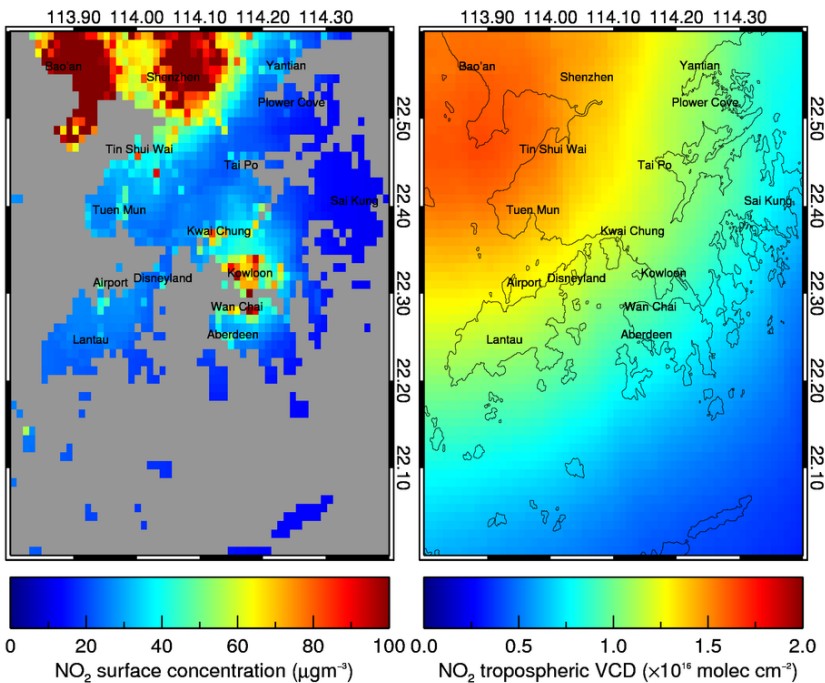

**Figure 2.** Comparison of the mean surface $NO_2$ concentration estimated by Model 1 (left) and the mean tropospheric VCD measured by OMI (right) during 2005-2015. Important areas are indicated.

Over the Hong Kong SAR, all models show clear enhancements over the areas already noted in Figure 2. The models also all predict a negative longitudinal gradient; concentrations predicted by the models over Lantau South Country Park (22.24° N, 113.93° E) were on average 2.6 times higher than those over Sai Kung Country Park (22.40° N, 114.35 ° E). This gradient may potentially be the result of in-situ station coverage; the most eastern station (Tap Mun) is situated in the Sai Kung Country Park, while the most western station (Tung Chung) is within a residential area and nearby Hong Kong International Airport.

The distribution of elevated $NO_2$ concentrations over the Hong Kong SAR does not significantly change between models, though the longitudinal gradient is more pronounced in some models than others. In Models 2-8 the gradient is strong enough to result in mean surface $NO_2$ concentrations predicted over Lantau South Country Park to be $\sim 40 \ \mu\mathrm{gm}^{-3}$. These values seem unrealistic, as the MODIS and WorldPop datasets suggest that the region is mostly uninhabited and undeveloped compared to districts like Aberdeen and Yantian, which show similar concentrations.

Outside of Hong Kong, the distribution of the Bao'an and Shenzhen enhancements change considerably between models, depending on whether road networks or population density and urban area coverage were used. Because of a lack of available surface concentration data from mainland China, these regions cannot be validated in this work.



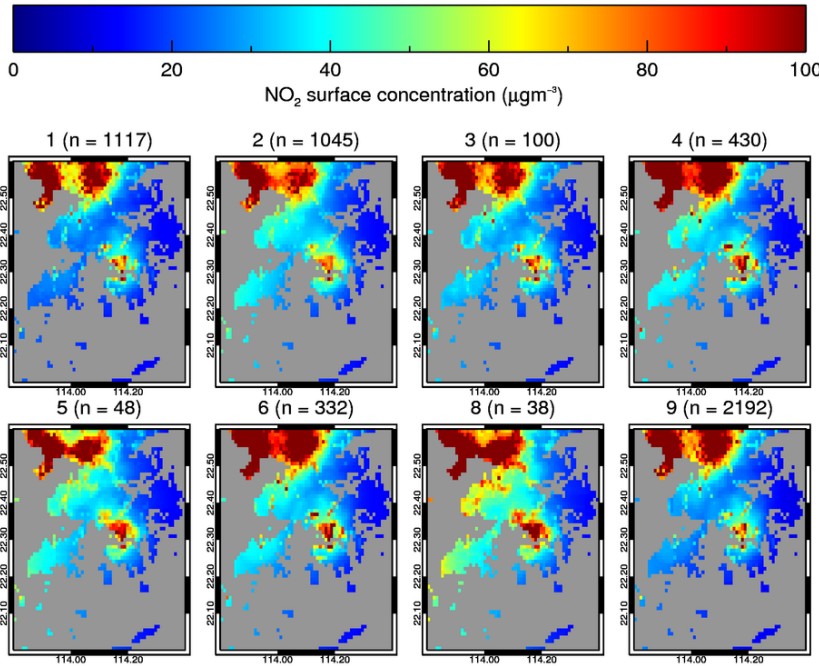

**Figure 3.** The mean surface $NO_2$ concentration predicted by each of the models listed in Table 2 for 2007-2012. Each plot also shows the number of cloud-free days from which the models could be trained.

## 3.2 Seasonal variation

All models including satellite data were found to predict higher surface $NO_2$ concentrations during the winter than in the summer, particularly over urban areas. This seasonal dependence may be caused by lower boundary layer height during winter, as well as increased emissions from residential heating. Figure 4 shows this seasonal gradient in the mean 2005-2015 predicted by Models 1 and 2 over both seasons.

Both models in Figure 4 are highly correlated in the summer ($R^2 = 0.97$), as they are largely based on the same variables, though Model 2 does not feature a longitudinal gradient. However, in winter the models are much less correlated ($R^2 = 0.78$), with Model 2 showing a much stronger longitudinal gradient than Model 1. As in Figure 3, this gradient leads to unphysically high concentrations being predicted over uninhabited regions such as Lantau South Country Park, making it unlikely that this is a realistic model of winter air quality over Hong Kong.

From Table 3 it is clear that the winter model had over 1000 fewer observations to use compared to the summer model. During winter there are fewer cloud-free observations, which would lead to the model overfitting the data available. Despite having fewer observations to use the winter model adjusted $R^2$ is higher than the summer model, suggesting that overfitting





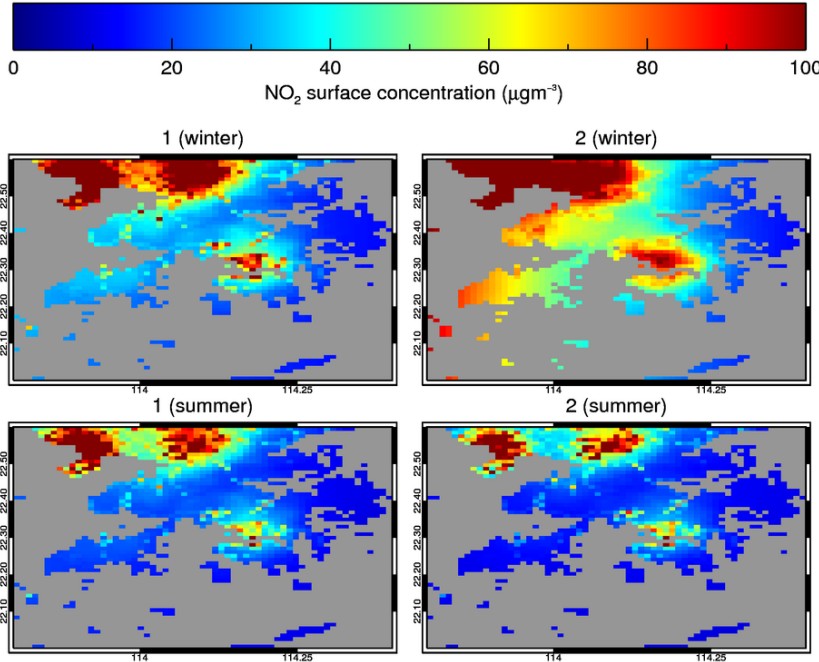

**Figure 4.** The mean surface $NO_2$ concentration predicted by Models 1 and 2 during winter (November-April) and summer (May-October) between 2005-2015.

has occurred. The spatial resolution of GOME-2 and SCIAMACHY are much larger than OMI, which would result in fewer cloud-free observations being available in the same time period, and so lead to the null results observed when seasonal models involving these datasets were attempted.

### 3.3 Cross-validation with in-situ data

5 Based on the adjusted $R^2$ values for each model shown in Table 3, it appears that Model 6 and 8 are the best performing models, suggesting that using more than one satellite dataset improves model prediction. However, the adjusted $R^2$ statistic may be artificially inflated by overfitting to the input data, and so may be overly optimistic descriptors of model performance.

Ideally these models would be validated against additional measured concentrations from stations independent of the current dataset. However, in the absence of other stations measuring ambient $NO_2$, the LUR models in this work were validated using

10 5-fold cross-validation (CV). In this process 80% of the measured data is used to calculate the coefficients and intercepts of each of the models shown in Table 2. These models are then used to estimate the surface concentrations of the remaining 20% of the data. This process is repeated until every data point has been estimated by a model that has not been trained using it.





| Model number | CV adjusted $R^2$ | CV gradient (error) | CV bias (error) | CV RMSE (%) |
|---|---|---|---|---|
| 1 | 0.775 | 0.889 (0.00376) | 5.14 (0.228) | 13.2 (24.4) |
| 2 | 0.838 | 0.840 (0.00290) | 7.23 (0.176) | 10.9 (20.1) |
| 3 | 0.745 | 0.865 (0.0170) | 7.24 (1.08) | 13.1 (22.4) |
| 4 | 0.808 | 0.844 (0.00670) | 7.67 (0.428) | 12.2 (21.1) |
| 5 | 0.586 | 0.861 (0.0420) | 9.25 (2.67) | 18.1 (40.0) |
| 6 | 0.480 | 0.583 (0.0104) | 20.9 (0.665) | 20.7 (36.1) |
| 8 | 0.535 | 0.990 (0.0629) | 1.89 (4.03) | 23.5 (40.0) |
| 9 | 0.419 | 0.447 (0.00266) | 25.6 (0.153) | 19.1 (36.9) |

**Table 4.** The results of the 5-fold cross-validation (CV) applied to all the LUR models described in Table 2. Surface concentrations estimated using CV were compared against the original in-situ measurements using linear regression, from which the adjusted $R^2$, gradient, bias, and RMSE ($\mu$gm$^{-3}$) are derived. The standard error of the gradient and bias are also displayed, while the RMSE is also expressed as a percentage of the mean concentration estimated by the model.

In this work the predictive performance of each model is determined through comparing the cross-validated model dataset against the original in-situ measurements through linear regression. Agreement between the two datasets is quantified by calculating the adjusted $R^2$, gradient, intercept, and root mean square error (RMSE, $\mu$gm$^{-3}$). The RMSE of a model was calculated as the square root of the mean of the squared errors. Table 4 shows the results of the cross-validation on each of the
models considered in this work.

From considering the CV adjusted $R^2$ and RMSE, it is clear that all the models including satellite data perform better than Model 9, suggesting that there is some utility in incorporating satellite data in LUR models. Model 2 has the highest CV adjusted $R^2$ and lowest RMSE, suggesting that OMI data offered the best agreement with in-situ measurements, so long as seasonal effects are accounted for. Sources of error reflected by the RMSE in Models 1-8 may be from coarse spatial sampling
by the satellite instrument, or retrieval algorithm errors in the satellite dataset.

Models using only one satellite dataset also perform better than those combining two or more datasets. A likely cause behind this difference is that the models using using more than one satellite dataset had fewer cloud-free observations to use, because of complications arising from different spatial resolutions and orbital coverage. A lack of available data would have therefore resulted in these models overfitting the input data available.

## 3.4 Spatial representivity

For all models in this work the CV dataset can be grouped by station, which allows for side-by-side comparisons of model performance over all regions to be made. Figure 5 shows the CV adjusted $R^2$ and RMSE for each model over each station. It is clear from the CV that with the exception of Tap Mun, Models 1-4 agree much better with the in-situ data overall compared to



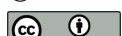

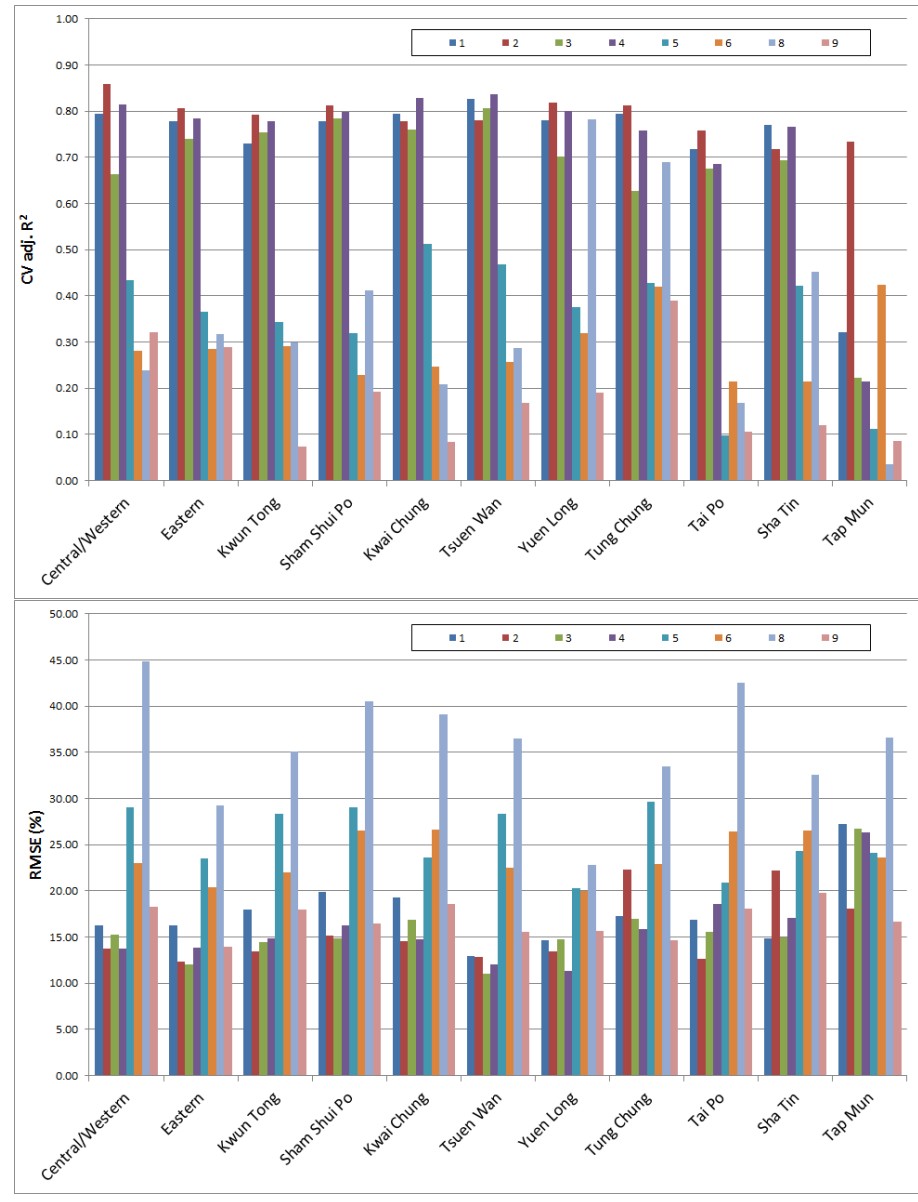

**Figure 5.** The CV adjusted $R^2$ and RMSE for each of the HK-AQN stations used in this work, as reported by the models listed in Table 2

Models 5-9. Figure 5 also shows that models 1-4 also on average have much lower RMSEs over most stations excluding Tap Mun, which suggests that they offer a higher precision than Models 5-9.





However, over Tap Mun almost all models (excluding Model 2) perform poorly, with lower adjusted $R^2$ values and RMSE values that are higher than the mean of the other stations. This result suggests that the models all have poor spatial representivity over unpopulated areas, which is because such regions are largely unrepresented by the in-situ stations.

Model 2 is somewhat of an outlier to this trend, as over Tap Mun the CV adjusted $R^2$ is 0.73, which is comparable to values
retrieved over the other stations. Similarly, the CV RMSE retrieved over Tap Mun is also lower than the value retrieved by Model 1. This suggests that training a season-specific model may better account for variability between rural and urban areas.

### 3.5 Temporal representivity

The CV datasets produced in this work can also be grouped and validated by year to determine whether annual or decadal changes in $NO_2$ are successfully predicted by models trained with all available data. The inclusion of satellite data as a
predictor variable also raises the possibility of instrument degradation affecting model performance. Unlike in-situ stations, satellite instruments can only be passively recalibrated over their lifetime, leading to a possible drift in retrieval precision that may progressively bias surface $NO_2$ models.

One example of this is the OMI row anomaly, which since 2007 has grown to affect half of the instrument orbital coverage. Over time, this would lead to fewer available observations, which may lead to biases in the LUR models. The degradation in
available measurements, combined with decreasing precision of the DOAS fit over time (Anand et al., 2015) should result in a decline in the annual CV adjusted R2 and a corresponding rise in the CV RMSE becaause of the increased uncertainty in the model.

Table 5 shows the annual CV adjusted $R^2$ and RMSE of Models 1 and 2 between 2005-2015. While no statistically significant trend is observed in the CV adjusted $R^2$ values for either model, both models show a statistically significant decline in RMSE
over time (Model 1: -0.28%yr$^{-1}$, Model 2: -0.11%yr$^{-1}$), which suggests that coverage losses or instrument degradation are not significant influences on model accuracy or precision. Table 5 also shows that on average the adjusted $R^2$ of Model 2 is $\sim 8.0\%$ higher than Model 1, while the RMSE is $\sim 23\%$ lower, suggesting that the better performance Model 2 showed in Table 5 was not the result of anomalously high correlation with in-situ measurements over certain years.

### 3.6 Influence of local meteorology

For every satellite dataset considered in this work, at no point in the model selection process were the meteorological variables from ERA-Interim considered for inclusion in the model, as other variables were found to improve the model adjusted $R^2$ more than adding temperature or wind speed. One possible reason for this may be that the spatial resolution of the ERA-Interim is too coarse to capture the true variation in temperature and wind speed. Another possibility is that the satellite data implicitly contain information about ambient atmospheric conditions observed as part of the VCD measurement, so meteorological data
may not be needed in the LUR model.

In order to determine whether meteorological data substantially improves the LUR model, Model 1 was trained again while forcing surface temperature and wind speed from ERA-Interim as predictor variables. The training process again selected the same variables shown in Table 3, with the addition of the total tertiary road length within 400 m. Wind speed and temper-





| Year | Model 1 CV adjusted $R^2$ | Model 2 CV adjusted $R^2$ | Model 1 CV RMSE (%) | Model 2 CV RMSE (%) |
|------|------|------|------|------|
| 2005 | 0.742 | 0.838 | 14.4 (26.0) | 10.2 (18.4) |
| 2006 | 0.744 | 0.822 | 14.2 (25.4) | 10.8 (19.3) |
| 2007 | 0.783 | 0.839 | 12.7 (23.7) | 9.86 (18.4) |
| 2008 | 0.804 | 0.849 | 12.6 (22.5) | 10.1 (17.9) |
| 2009 | 0.779 | 0.840 | 12.4 (23.2) | 9.80 (18.3) |
| 2010 | 0.788 | 0.848 | 12.0 (22.8) | 9.27 (17.6) |
| 2011 | 0.793 | 0.841 | 11.9 (21.6) | 9.58 (17.4) |
| 2012 | 0.744 | 0.807 | 11.9 (22.2) | 9.44 (17.7) |
| 2013 | 0.791 | 0.845 | 13.3 (23.2) | 10.2 (17.8) |
| 2014 | 0.785 | 0.849 | 11.7 (23.6) | 8.93 (18.1) |

**Table 5.** The adjusted $R^2$ and RMSE ($\mu gm^{-3}$) determined from the 5-fold cross-validation (CV) applied to Models 1 and 2 (see Tables 2 and 4), grouped by year.

ature were found to have a negative effect on surface concentration; the ERA-Interim temperature may represent the rate of photochemical dissociation of $NO_2$, while high wind speeds would increase mixing and therefore act to lower concentrations. Figure 6 shows the seasonal average surface $NO_2$ concentration predicted by Model 1 with and without meteorological data for 2005-2015. The addition of meteorological data causes a $\sim 17\%$ mean increase in surface $NO_2$ concentrations across the region, though no new emission sources are visible.

As with the other models, this model variant can be validated against the in-situ measurement data using 5-fold CV and compared with the results in Table 4. When meteorological data was forced the CV adjusted $R^2$ was 0.806, compared with 0.775 before, suggesting that the inclusion improves the model agreement. Similarly, the model CV RMSE decreased to 12.0 $\mu gm^{-3}$ (22.1%) after including meterological data. The CV gradient also decreased to 0.846, while the CV bias became 7.17 $\mu gm^{-3}$. The decrease in gradient and increase in bias suggests that the inclusion of ERA-Interim data does not improve the LUR model accuracy, though the increase in CV adjusted $R^2$ and decrease in RMSE shows that it does improve the precision of the model.

For this work it is thought that the effect of meteorological data in the LUR model is limited by the spatial resolution of the satellite instruments, or the ERA-Interim dataset. Previous LUR models incorporating daily meteorological data (e.g. Su et al., 2008; Lee and Koutrakis, 2014) have typically used measurements from weather stations either close to or at the sites where the $NO_2$ concentrations have been measured, with the ambient temperature and wind field therefore interpolated from these fixed points. Because of the comparatively fewer number of $NO_2$ stations available for this work, it was thought that a harmonised dataset like ERA-Interim would reduce the spatial uncertainty otherwise introduced by discrete weather stations. Future iterations of this work should investigate if using in-situ weather data would provide a better outcome.



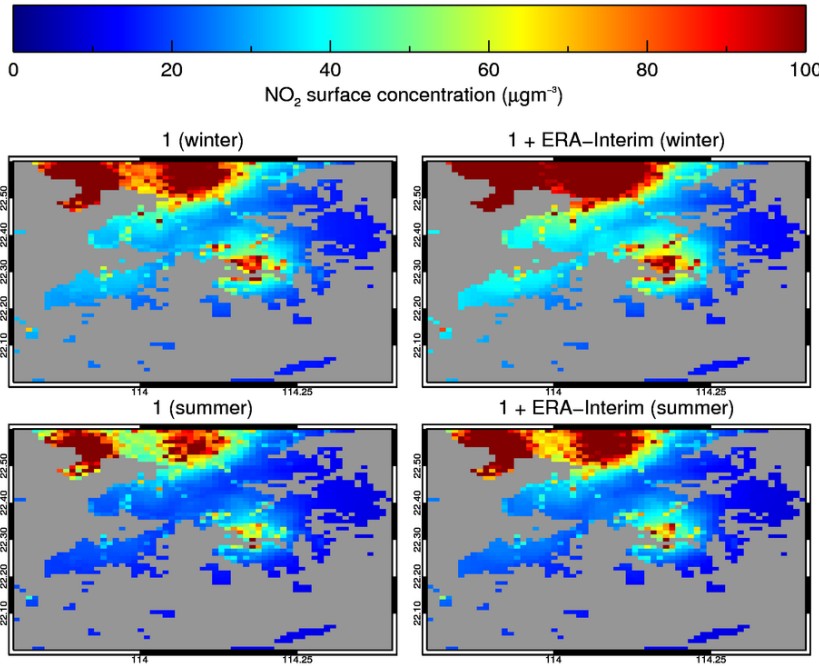

**Figure 6.** The mean surface $NO_2$ concentration predicted by Model 1 and 2 during winter (November-April) and summer (May-October) between 2005-2015, with and without the inclusion of wind speed and temperature from the ERA-Interim reanalysis dataset (Dee et al., 2011).

### 3.7 Validation using OMI and MACC-II reanalysis data

An alternative technique to deriving surface $NO_2$ concentrations from satellite measurements is to use a chemical transport model to estimate the vertical profile at the time of the satellite overpass (Lamsal et al., 2008). The profile can then be used to partition the tropospheric VCD into its surface and free-tropospheric components, thereby estimating a scaling factor that
5   can be applied to the measured VCDs. This approach is advantageous in that it allows for surface $NO_2$ concentrations to be mapped at a higher spatial resolution than many CTM grids.

For this work a similar approach to Lamsal et al. (2008) was used to infer surface $NO_2$ concentrations from OMI data. Daily mean $NO_2$ vertical profiles over Hong Kong were sampled from the MACC-II reanalysis dataset (Monitoring Atmospheric Composition and Climate, Inness et al., 2013) for this purpose. For an OMI ground pixel, $O$, the surface $NO_2$ concentration,
10   $S_O$, is estimated from the OMI tropospheric VCD, $\Omega_O$, using the following relation:

$$S_O = \frac{\nu S_G}{\nu \Omega_G - (\nu - 1)\Omega_G^F} \times \Omega_O \qquad (3)$$


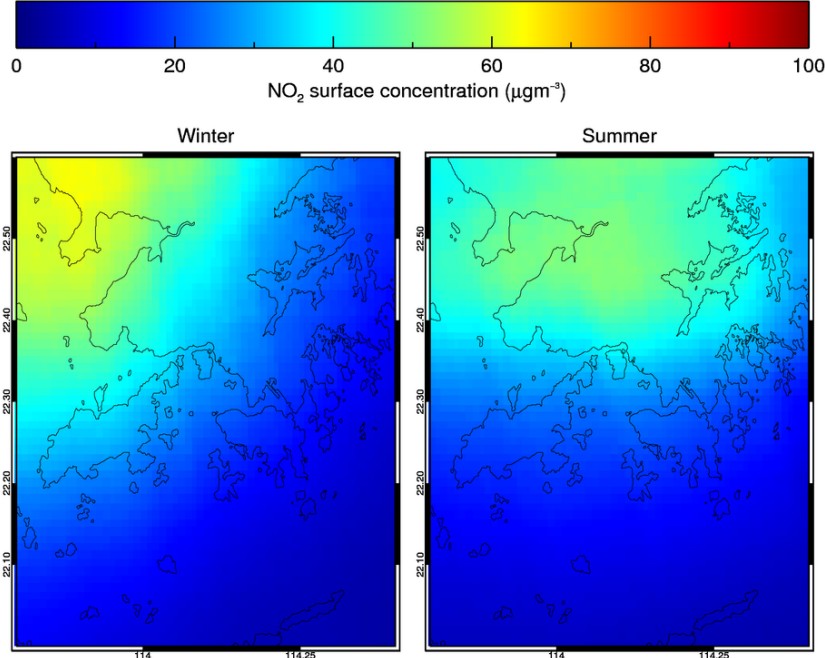

**Figure 7.** The mean surface NO$_2$ concentration inferred from OMI tropospheric VCDs using MACC-II reanalysis data, between 2005-2012. Data is plotted for winter (left, November-April) and summer (right, May-October).

Here, the terms $\Omega_G$ and $S_G$ are the tropospheric VCD and the surface concentration derived from the MACC-II daily average profile, for which the surface is defined as the lowest layer of the profile (20 m). To obtain the tropospheric VCD the profile is integrated up to the tropopause height taken from the OMNO2 dataset. The modelled free tropospheric NO$_2$ column, $\Omega_G^F$, is taken to be horizontally invariant over the MACC-II grid cell, in order to represent the longer NO$_x$ lifetime in the free

5   troposphere. As the spatial resolution of the MACC-II dataset is much larger than the OMI nadir resolution ($1.125° \times 1.125°$), the $S/\Omega$ conversion factor is weighted by an additional term, $\nu$, which is defined as the ratio of the local OMI tropospheric VCD to the mean OMI field over the MACC-II grid cell.

MACC-inferred surface concentrations were calculated for all cloud-free OMI pixels measured over Hong Kong between 2005-2012 and compared against the daily ambient NO$_2$ concentrations recorded at the in-situ stations. Figure 7 shows the

10   mean surface NO$_2$ concentration estimated using MACC-II and OMI data for winter and summer over Hong Kong. Compared to Figure 4, it is clear that the MACC-inferred concentrations are much lower and capture much less spatial information than the LUR models, because of limitations placed by the OMI spatial resolution. Over both seasons, NO$_2$ concentrations appear to peak north of the Hong Kong SAR, potentially caused by emissions from Shenzhen and Bao'an, or transported further north from the Pearl River Delta.



Because of this lack of spatial detail, the MACC-II concentrations correlate very poorly with the in-situ data ($R^2 = 0.11$, RMSE= $41.9 \mu \text{gm}^{-3}$), with a linear gradient of $\sim 0.58$. As well as this, previous comparisons of tropospheric $NO_2$ VCDs inferred from MACC-II profiles with SCIAMACHY data over East Asia suggest that the dataset underestimates tropospheric $NO_2$ by a factor of two in winter (Inness et al., 2013), which may also partially explain the lack of agreement with the in-situ

data. It is clear from this result that the mixed effects LUR model offers better spatial resolution and predictive capability than the MACC-II reanalysis over Hong Kong.

### 3.8   Time series analysis

The Model 1 dataset covers a decade of near-continuous measurements, from which it may be possible to determine whether $NO_2$ concentrations have significantly changed after accounting for noise and seasonal variation. To determine if a statistically

significant trend can be observed from this dataset, surface concentrations modelled over Kowloon and Hong Kong Island (see Figure 1) were binned to monthly averages between 2005-2015. Following Hilboll et al. (2013), a linear trend with a seasonal component was fitted to this time series. The surface concentration at month $t$ ($Y(t)$, where $t = 0$ is January 2005), was modelled as a combination of a fixed intercept $\mu$ and linear trend $\omega$:

$$Y(t) = \mu + \omega t + (1 + \xi) \times \sum_{j=1}^{4} \left( \beta_{1,j} \sin\left( \frac{2\pi j t}{12} \right) + \beta_{2,j} \cos\left( \frac{2\pi j t}{12} \right) \right) + N(t) \tag{4}$$

The time series may be subject to changes in the seasonal component caused by the relative disparity between constant and seasonal emissions (e.g. transport and heating). To reflect this, an additional term, $\xi$ is introduced to equation (4) to dampen or drive the seasonal oscillation over time. The term $N(t)$ represents the noise component (i.e. the remaining signal in the time series that cannot be explained by the model)

Equation (4) is first solved using nonlinear regression to determine the values of $\mu$, $\omega$ and $\xi$ that minimise $N(t)$. The seasonal

components have a negligible impact on the estimation of the other parameters in equation (4) (Weatherhead et al., 1998), so these are subtracted from the time series. In addition to this, the autocorrelations are also accounted for using a linear matrix transformation. Finally, linear regression is applied to determine $\mu$ and $\omega$ (Mieruch et al., 2008).

In order to determine the linear trend error, it is assumed that the noise $N(t)$ is autoregressive with lag 1 (AR(1)). Following the approach defined by Mieruch et al. (2008), the linear trend is considered to be statistically significant only if the following

condition is satisfied:

$$P_{H_0}\left( |\hat{\omega}| > 2\sigma_{\hat{\omega}} \right) = \text{erf}\left( \frac{|\hat{\omega}|}{\sigma_{\hat{\omega}}\sqrt{2}} \right) > 95\% \tag{5}$$

where $\text{erf}(x)$ is the Gauss error function.

The monthly average time series and the fitted model are shown in Figure 8, along with an annual bottom-up $NO_x$ emission inventory estimated by the HKEPD (HKEPD, 2014). The linear trend was estimated to be: -0.0208 $\mu \text{gm}^{-3}\text{yr}^{-1}$ ($-0.430\% \text{yr}^{-1}$




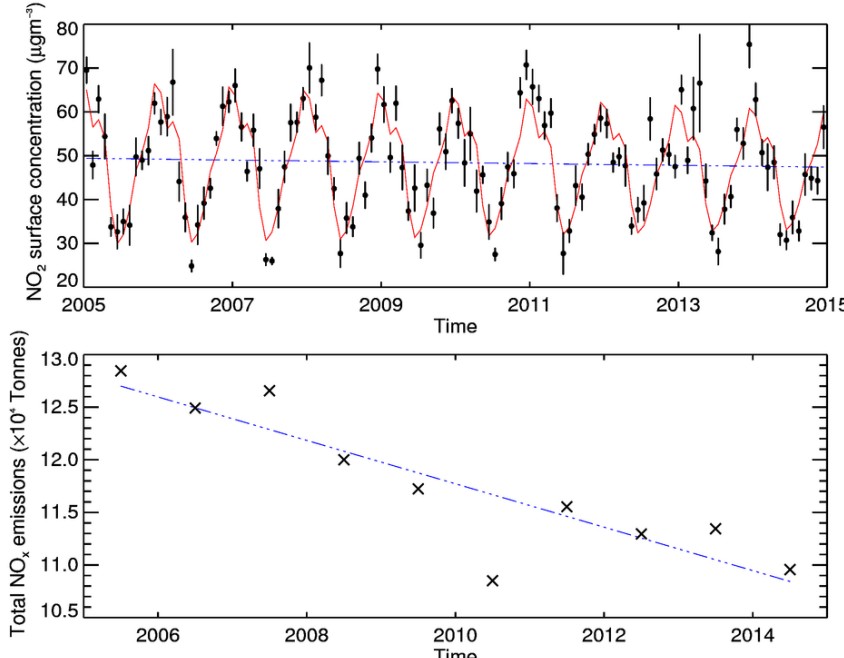

**Figure 8.** (top) Time series analysis of the monthly mean surface $NO_2$ between 2005-2015 predicted by Model 1 (see Table 2) over the region covering Kowloon and Hong Kong Island shown in Figure 1. The error bars represent the standard error of the mean for each month, while the red line represents the linear trend and seasonal cycle modelled using equation (4). The linear trend is also shown separately as the blue dashed line. (below) The annual total $NO_x$ emissions by Hong Kong, as estimated by the HKEPD bottom-up inventory (HKEPD, 2014).

relative to the average 2005 concentration). The seasonal dampening term $\xi$ was estimated to be: -0.0287 $\mu\mathrm{gm}^{-3}\mathrm{yr}^{-1}$. However, the trend was found to be statistically insignificant. A similar result was found when analysing satellite data between 1996-2012 over Hong Kong by Hilboll et al. (2013), who also found that the signs of $\mu$ and $\xi$ were the same. Another investigation by Schneider et al. (2015) using only SCIAMACHY data also found a statistically insignificant negative trend, as well

5  as a statistically significant trend of $-3.8\%\mathrm{yr}^{-1}$ over Shenzhen.

A statistically insignificant negative trend was also estimated when this analysis was repeated using data predicted by Model 2 ($-0.537\%\mathrm{yr}^{-1}$), as well as the spatial mean concentration reported by the in-situ stations in this region ($-0.240\%\mathrm{yr}^{-1}$). By contrast, the HKEPD inventory shows a statistically significant trend of $-1.60\%\mathrm{yr}^{-1}$. A possible reason behind this discrepancy could be influence from $NO_x$ emissions transported from mainland China which may obscure any decline in local

10  emissions. The OMI spatial resolution may also affect the precision of tropospheric VCDs measured over Hong Kong, which may result in a negative bias in surface concentrations modelled.





## 4 Conclusions

The Hong Kong SAR is subject to high ambient $NO_2$ concentrations caused by a combination of local emissions and pollution transported from elsewhere in the Pearl River Delta. Exposure studies require the calculation of accurate surface concentration maps, which could be enhanced by the synoptic coverage offered by satellite instruments. For this work several mixed effects LUR models were developed to explore this concept, which combined in-situ $NO_2$ measurements with tropospheric VCDs measured by satellite instruments. Compared to a reference LUR model with no satellite data or mixed effects, all models were found to have superior predictive performance in estimating daily ambient $NO_2$ concentrations over the region, with an average CV adjusted $R^2$ of 0.681.

The LUR models used high spatial resolution datasets such as road networks and MODIS land cover to simulate likely emission sources. This allowed for distinct features to be visible over districts such as Kowloon, Yantian, and Wan Chai ($\sim 100 \mu gm^{-3}$). By contrast, local minima were observed over uninhabited areas such as the Sai Kung and Plower Cove Country Parks ($\sim 5 \mu gm^{-3}$). One anomaly to this trend was the Lantau South Country Park, which was modelled to have ambient $NO_2$ concentrations as high as $40 \mu gm^{-3}$. This enhancement may be the result of pollution from the nearby Hong Kong International Airport, or an artefact caused by the location of the Tung Chung station. Very large features were also observed over Shenzhen and Bao'an, though validating these are beyond the scope of this work due to insufficient station coverage.

For this work several models were developed to assess the relative utility of OMI, SCIAMACHY, and GOME-2 data as predictor variables. The quality of these datasets differs significantly because of their temporal sampling and spatial resolution. From 5-fold cross-validation with the in-situ data it was found that OMI data gave the best agreement with the in-situ data, so long as seasonal effects were accounted for (CV adjusted $R^2 = 0.838$). OMI has the smallest spatial resolution and the longest temporal range of the three instruments, which allowed for local emissions and the seasonal cycle to be better accounted for. Larger ground pixel sizes are at risk of contamination by pollution transported from Shenzhen or elsewhere in the PRD, which may add a positive bias to the inferred surface concentrations.

It was thought that the models including more than one satellite dataset would have improved sensitivity to diurnal variation, and so perform better than those with single dataset. However, as with all statistical models, the LUR model performance is dependent on the number of observations available. As only cloud-free satellite data can be used, the number of available observations is therefore heavily dependent on the season and the spatial resolution of the satellite instrument (Krijger et al., 2007). Factoring diurnal changes in cloud cover, this means that models using more than one satellite instrument would be fitted using fewer observations than single instrument models. Because of these issues and differences in spatial resolution, it was difficult to determine whether diurnal cycle coverage was accounted for by these models.

By collating cross-validation model data by in-situ station and time it was possible to gauge the spatiotemporal representivity of each model. For models using only OMI data no significant negative trend in the CV adjusted $R^2$ was found between 2005-2015, suggesting that these models can account for the progressive loss of coverage caused by the row anomaly, allowing for high temporal representivity over the entire observation period.





The single-instrument models generally performed better than the multiple-instrument and reference models over all regions except for the rural Tap Mun station, where all models apart from the seasonal OMI model performed poorly. Tap Mun is the only rural station in the HK-AQN, which may have resulted in the models being biased in favour of highly polluting urban areas.

One example of this bias is the longitudinal gradient present in most of the models, which is especially notable in Figure 4. The longitudinal gradient has resulted in unrealistically high concentrations being reported over the uninhabited Lantau South Country Park, which raises concerns over the true spatial representivity of the models over regions where no in-situ data is available. Future iterations of this work may require a more diverse in-situ network and/or higher resoluton satellite data to better capture the spatial gradient between polluted and unpolluted regions.

For this work temperature and wind information from the ERA-Interim reanalyis dataset was provided in the model training process, in order to simulate photochemical loss and mixing. However, at no point in the training process for any of the models including satellite data were these variables considered for inclusion, as other variables were found to improve the model performance more. When temperature and wind speed were forced into Model 1, the average $NO_2$ concentration over the region increased by $\sim 17\%$, though no new features were observed. Cross-validation with the in-situ data suggests that while including ERA-Interim data improves model precision, the model accuracy falls. One possible cause of this decrease in accuracy may be that the spatial resolution of ERA-Interim was too coarse to fully represent the true atmospheric state. The model performance may potentially be improved if in-situ measurements from a dense network of weather stations could be used instead.

Time series analysis was applied to surface concentrations predicted by the OMI-only models to determine whether a trend in emissions over Kowloon and Hong Kong Island could be determined between 2005-2015. Both models and the in-situ data over this region reported a statsitically insignificant trend over this region ($-0.430\%\mathrm{yr}^{-1}$ for Model 1). By contrast, the HKEPD annual bottom-up $NO_x$ inventory suggests that a statistically significant trend of $-1.60\%\mathrm{yr}^{-1}$ should be observed during this period. Emissions transported from elsewhere in the PRD may have offset any observable decline in local emissions, though this would require accurate information of pollution outside of Hong Kong to verify.

In the absence of additional in-situ data surface $NO_2$ concentrations were also estimated from OMI data using profiles from the MACC-II reanalysis dataset. However, surface concentration maps derived using this method had the same spatial resolution as OMI, and so were dominated by pollution transported from Shenzhen or further afield. As well as this, the MACC-II dataset has previously been shown to have poor agreement with other satellite datasets over East Asia (Inness et al., 2013), which may also affect the accuracy of this method. Because of these issues, agreement with in-situ data was poor ($R^2 = 0.111$) compared with the models used in this work. It is likely that better estimates could have been achieved with higher spatial resolution CTMs, such as the Models-3 Community Multiscale Air Quality (CMAQ, Kuhlmann et al., 2015).

This work has demonstrated the potential in combining in-situ data with satellite data with a mixed effects model to obtain better estimates of daily surface $NO_2$ concentrations over Hong Kong, and can be applied to other megacities in China and elsewhere, so long as a diverse in-situ monitoring network exists to calibrate and validate the model. However, the spatial resolution of the satellite instrument remains a source of error, which may lead to underestimating the true surface concentration





over megacities. In the future, the performance of this model would be greatly improved by the inclusion of higher resolution satellite data from forthcoming missions such as Sentinel-5P ($7 \times 7$ km, Veefkind et al., 2012). Accounting for diurnal cycle

variability in daily estimates may also still be possible by combining daily measurements made by instruments with similar spatial resolutions (e.g. Geostationary Environment Spectrometer, GEMS, Kim, 2012).

*Acknowledgements.* The research leading to these results has received funding from the European Union Seventh Framework Programme ([FP7/2007-2013]) under grant agreement no. 606719, as part of the PArtnership with ChiNa on space DAta (PANDA) project. Additional funding was also provided by the UK National Environmental Research Council (NERC) under grant no. NE/N006941/1, as part of An

Integrated Study of AIR Pollution PROcesses in Beijing (AIRPRO).

We acknowledge the use of OMI data made available from the NASA MIRADOR service(http://disc.sci.gsfc.nasa.gov/Aura/data-holdings/OMI), as well as the use of SCIAMACHY and GOME-2 data provided by the KNMI TEMIS (http://www.temis.nl) service. The ERA-Interim and MACC-II reanalysis datasets were provided by ECMWF (http://www.ecmwf.int). The in-situ $NO_2$ measurements and $NO_x$ emission inventory were provided by the Hong Kong Environmental Protection Department (http://www.epd.gov.hk/epd/eindex.html). OMI data gridding

was made possible using software kindly provided by Dr Gerrit Kuhlmann, available at: https://github.com/gkuhl. This research used the SPECTRE High Performance Computing Facility at the University of Leicester.





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
