# Peer review of "Estimating daily surface NO2 concentrations from satellite data - A case study over Hong Kong using land use regression models"

_Atmospheric Chemistry and Physics, 2016_

## Referee Comment (RC1) · Anonymous Referee #1 · 16 Feb 2017

This study presents development of a mixed-effect LUR model to estimate surface NO2 concentrations over the Hong Kong SAR. In-situ measurements from surface network and tropospheric NO2 column data from multiple satellite instruments are combined with fine-scale land use parameters to predict daily surface concentrations. Their analysis shows that models with satellite data and mixed-effect LUR show superior predictive performance as compared to the reference LUR model. Similar conclusions were drawn by few previous studies, albeit different regions. This study goes beyond other studies by exploring the models' predictive ability with data from multiple satellite instruments. The manuscript is well written. However, I have few concerns as listed below that need to be addressed before it can be published in ACP.

General comments

1) Some results may point to deficiency in method or errors in data analysis. First, the authors state in Section 3 that models combining OMI and SCIAMACHY data always failed to converge, which suggests a problem in their implementation. Second, the model (and their interpretation) seems to neglect some important predictive parameters such as NOx lifetime. Third, average NO2 concentrations presented in Figure 7 are not consistent with seasonal behavior of NO2 (peaking in winter time) especially over regions east of 114 deg longitude. Fourth, their estimated trend contradicts results from several other trend studies over Hong Kong and is not consistent with the trend in emissions.

2) The work is built on a poor foundation. The authors use satellite data obtained from different sources. As a result, retrievals are not consistent due to differences in all aspects of retrieval algorithm – from spectral fit to the use of various input parameters. The first task should have been checking consistency between different data set. Assuming each data product as truth is another major limitation. Therefore, it might be more helpful to focus on measurements from a single instrument and carry out a thorough investigation rather than presenting lengthy and speculative discussions.

Specific comments

Page 4, lines 2-3: This sentence is incomplete, please revise.

Page 4, line 14-15: Reverse the citation.

Page 5, lines 24: What do you mean by NO2-Omega relationship? Please, clarify.

Page 6, line 19: How does ocean deposition affect local NO2 concentration? Describe the mechanisms if that is indeed the case.

Page 7, lines: 5-6: Correct R2 here and in other places.

Page 9, lines 7-8: You state "This null result may be due to a lack of cloud-free days

when both instruments were coincident over Hong Kong." This may point to some deficiency in your implementation. Why is it necessary to have cloud-free observations for both instruments?

Page 10, line 3: I do not understand your statement "vertical mixing being dominated by emissions from mainland China." How would distant sources affect vertical mixing?

Page 11, Figure 2: What does the gray area represent? What does the data gap in the mean surface $NO_2$ map mean? What explains the large spatial gradient (box-to-box gradient) in the mean concentration map? Wouldn't wind transport pollution to neighboring areas?

Page 12, lines 3-4: The wintertime enhancement is also due to increase in $NO_x$ lifetime in winter.

Page 13, line 1: Your statement "The spatial resolution of GOME-2 and SCIAMACHY are much larger than OMI" is not correct. OMI has higher spatial resolution than GOME-2 and SCIAMACHY.

Page 14, lines 11-14: It is pity that you are not recognizing the fact that there is large inconsistency in retrievals. Please, see my general comments.

Page 16, lines 9-12 and lines 14-17: Are there any studies that suggest effect of instrument degradation in satellite $NO_2$ retrievals? I would be surprised if DOAS-type retrievals from satellite can have significant impact from instrument degradation.

Page 16, line 13: I believe, the terminologies "row anomaly" and "instrument degradation" are not same. Data affected by row anomaly are not supposed to be used.

Page 16, lines 20-21: Wouldn't your statement "which suggests that coverage losses or instrument degradation are not significant influences on model accuracy or precision" here and in other places contradict your discussions regarding SCIAMACHY (less sampling due to global coverage in 6 days) and GOME-2 (more cloudy pixels)?

Page 16, lines 25-27: Don't understand this, suggest revise the statement.

Page 17, lines 1-2: I wonder how temperature can be a proxy for photochemical dissociation of NO2. Shouldn't it be actinic flux?

Page 17, lines 10-11: What is your measure for your model accuracy? Why are improvement in R2 and decrease in RMSE not considered for model improvement?

Page 19, line 1: What is the logic behind applying daily average profiles instead of early-afternoon profiles that are more relevant for OMI? Could this be the reason for low correlation between OMI and in-situ observation?

Page 20, line 15: Seasonal variation is driven mostly by changes in NOx lifetime and emissions.

Page 21, Figure 8: Deviation of red curve (fitted line) considerably from data points may suggest that the term in Eqn 4 that accounts for seasonal variation over time may not have been properly applied. Visually, the area under the curve passing through the points seems decreasing over time, consistent with the trend in emissions. Please check your calculation of trend. Please show the trend in OMI column data as well.

Page 21, lines 10-11: Clarify this statement. How does the change in precision result in negative bias in surface concentration?

Page 22, line 20: Your statement "OMI has the smallest spatial resolution" is incorrect. It should have been either highest spatial resolution or smallest pixel size.

Page 22, line 23: Where is that positive bias - background regions, polluted areas, or everywhere?

Page 22, line 24: Why would multiple satellite data have improved sensitivity? Sensitivity to diurnal variation? It could provide information on diurnal changes, only if the measurements are self-consistent (instrument, algorithm, etc.).

Page 22, lines 28-29: I don't understand this. Why do you need to exclude clear-sky

OMI observations if SCIAMACHY observations are cloudy for a given day? This might point to deficiency in your approach.

Page 23, lines 13-15: Unclear. Revise this statement.

Page 24, line 6: Correction: Geostationary Environmental Monitoring Spectrometer.

———————————————

---

## Referee Comment (RC2) · Anonymous Referee #2 · 28 Mar 2017

In this manuscript, Anand and Monks present the development of a land use regression model to predict daily surface NO2 concentrations over Hong Kong by combining satellite observations of NO2 tropospheric columns with finer scale land use parameters. This approach infers high-resolution spatial features in NO2 surface concentration using land use parameters while incorporating the long-term synoptic temporal coverage captured by satellite measurements. They show that their land use regression model predicts surface NO2 concentrations more successfully than estimates obtained by combining satellite observations with modeled NO2 vertical profiles. The lack of a trend in predicted surface NO2 concentrations contradicts expectations from a bottom-up NOx inventory for the region which shows a significant declining trend.

[Figure]

My main concern with this manuscript is one of scope. This manuscript is primarily of local and technical interest. While air quality in this region is certainly of interest to the community, I'm not convinced that the authors have developed the land use regression approach sufficiently for it to have novel general implications. The authors acknowledge their model is similar to a previously developed mixed-effects model, and thus the advancement here is largely in its application to a different region. If the authors were presenting a radical new approach to land use regression, then the regional nature of the study would be less of an issue. As it stands, I wonder whether ACP is the appropriate journal for publication. I urge the authors to present a clear argument for how this paper makes a more general contribution to atmospheric science.

Having said that, the study is interesting, and the manuscript is written clearly. If publication in ACP is to be pursued, I have the following general and specific comments to take into consideration.

General comments:

I am not yet satisfied that 11 monitoring stations are sufficient to build a national-scale LUR model. As they admit, the coverage is predominantly urban. We have no sense of what "dynamic range" exists in the predictor variables across these urban sites, and whether there is enough to build a robust model. My problem is that while there are many individual daily observations, parameters like road length, urban area coverage, population density, vegetation area coverage, and elevation will not vary day-to-day. Thus, they only have 11 unique data points for each variable. This strikes me as an extremely small sample. One transparent way they could address this is by including regression plot results for each predictor in a supplementary section.

Another test would be to try building the model(s) by holding back specific ground-station locations (and not just a percentage of the available data from each location, since this does not test the sensitivity to the loss of a ground station location). Was this the purpose of Section 3.4? I was a bit confused about what this section was

presenting. In general, I am not satisfied that a cross-validation approach involving removing 20% of all the data is a good enough test. It strikes me that a better test would be to remove entire stations instead. Keeping any observations from every station basically means that most of the predictor variables aren't actually getting tested (since these are not changing day-to-day).

Finally, I think the introduction to Section 2.3 about mixed effects LUR could use more detail. What would the formulation of a "fixed"/"random" effects model look like in comparison? What is the advance in the mixed effects model? As someone who is not overly familiar with the LUR approach, it is not really clear to me what the mixed effects model actually captures and how it is calculated (despite the inclusion and explanation of Equation 1). The authors should assume that some of the ACP audience will not be familiar with LUR.

Specific comments:

Section 2.1: Can you describe in a more detail the network of DOAS vs. in-situ measurements? How many of each? Do they have the same length of coverage? Are the data for both available online? Can you show which ones are which on the map? Are the chemiluminescence measurements via catalyst or LED conversion?

Equation 1: Should the omega symbol have both a subscript i and subscript j?

---

## Author Comment (AC1) · 13 May 2017

We thank the reviewer for their valuable comments. We have amended the manuscript in light of their suggestions, and will address the most pressing of these herein.

**General Comments**

*Some results may point to deficiency in method or errors in data analysis. First, the authors state in Section 3 that models combining OMI and SCIAMACHY data always failed to converge, which suggests a problem in their implementation. Second, the model (and their interpretation) seems to neglect some important predictive parameters such as NOx lifetime. Third, average NO2 concentrations presented in Figure 7 are not consistent with seasonal behavior of NO2 (peaking in winter time) especially over regions east of 114 deg longitude. Fourth, their estimated trend contradicts results from several other trend studies over Hong Kong and is not consistent with the trend in emissions.*

We shall respond to each point raised here in turn:

- Efforts to combine SCIAMACHY and OMI measurements with the mixed effects model are severely hampered by the significant difference in their spatiotemporal resolution (SCIAMACHY achieved global coverage only every 6 days with a spatial resolution of 60 x 30 $km^2$, while OMI achieves daily global coverage with a nadir spatial resolution of 13 x 24 $km^2$) and because no SCIAMACHY data exists after April 2012. Because of the additional criteria of having cloud-free measurements from both instruments on the same day, models including SCIAMACHY data used much fewer data points than the other models, as shown in Table 3. Because of these factors, it is possible that the resulting satellite datasets did not show a significant correlation with the daily in-situ data or the other predictor variables, and so a reliable model fit could not be obtained.
- Previous Land Use Regression studies have often been used to map the average spatial distribution of $NO_2$ over long time periods, and so did not need to factor the variable lifetime of $NO_x$ in their formulation. From our understanding there is no physical proxy that can fully replicate these variables, so for this work we assumed that properties such as the $NO_x$ lifetime were implicitly represented in the daily satellite data and the S/Ω relationship derived using the in-situ data.
- The data presented in Figure 7 was derived from MACC and OMI data. Analysis of OMI data over the Pearl River Delta has previously shown that tropospheric $NO_2$ VCDs peak in China during the winter, in line with increased anthropogenic emissions from heating (Wang et al, 2015). Similarly, tropospheric $NO_2$ VCDs derived from the MACC reanalysis product also show a similar peak during winter (Inness et al, 2013). We are unaware of any studies that contradict this, and would welcome further input on this matter.
- As stated in the manuscript, we believe that emissions from mainland China may be masking any local decline in emissions in our model. Contrary to Hong Kong, a number of $NO_x$ emission inventories show that mainland China emissions continuously increased for much of the 2005-2015 period (e.g. Ding et al, 2017), and so may have offset any decline in local emissions. The lack of a statistically significant $NO_2$ trend over Hong Kong has previously been shown in analyses of satellite data (Hilboll et al, 2013; Schneider et al, 2015)..

*The work is built on a poor foundation. The authors use satellite data obtained from different sources. As a result, retrievals are not consistent due to differences in all aspects of retrieval algorithm – from spectral fit to the use of various input parameters. The first task should have been checking consistency between different data set. Assuming each data product as truth is another major limitation. Therefore, it might be more helpful to focus on measurements from a single instrument and carry out a thorough investigation rather than presenting lengthy and speculative discussions.*

Data from these instruments over East Asia were previously compared with ground based MAX-DOAS measurements by Irie et al (2012), who concluded that the effective differences between the various retrieval algorithms were small and insignificant. We have since added this citation to the manuscript. In addition to this, data from GOME-2 and SCIAMACHY were retrieved using the same retrieval algorithm (TEMIS TM4NO2A), which should further minimise potential biases between these instruments. Single instrument models were also developed for this work, which were assessed in turn using cross-validation with the in-situ data in Section 3.3 and 3.4.

*How does ocean deposition affect local $NO_2$ concentration? Describe the mechanisms if that is indeed the case*

In previous studies (e.g. Ross et al, 2006) the distance to the ocean was introduced to LUR models to model the marine background $NO_x$ concentration. Depending on the wind direction, the coastal regions in Hong Kong may receive cleaner air from the South China Sea, and so would have lower $NO_2$ concentrations. We have altered the manuscript to replace this incorrect statement.

*Why is it necessary to have cloud-free observations for both instruments?*

As stated in the manuscript, the models developed in this work aim to predict the daily $NO_2$ concentrations using purely empirical data. Models using data from more than one satellite instrument aim to account for the diurnal cycle on that day. Therefore, measurements from both instruments within cloud-free parameters are required.

*I do not understand your statement "vertical mixing being dominated by emissions from mainland China." How would distant sources affect vertical mixing?*

We had meant to say here that OMI observations of $NO_2$ over Hong Kong will be dominated by the comparatively higher emissions from Shenzhen and Bao'an. We have replaced this statement in the manuscript.

*Figure 2: What does the gray area represent? What does the data gap in the mean surface NO2 map mean? What explains the large spatial gradient (box-tobox gradient) in the mean concentration map? Wouldn't wind transport pollution to neighboring areas?*

The grey area represents regions beyond the scope of the model – oceans and areas where no cloud-free satellite measurements were available. As stated on Page 9, the spatial gradients present in the concentration map are because of spatial gradients in emission sources (i.e. densely populated areas). The maps of modelled concentrations in this work are of temporally averaged data, so transport due to wind advection are not visible.

*Are there any studies that suggest effect of instrument degradation in satellite NO2 retrievals? I would be surprised if DOAS-type retrievals from satellite can have significant impact from instrument degradation.*

For OMI, we are unaware of such studies outside of Anand et al (2015), which showed a gradual decrease in the precision of the fitted total $NO_2$ slant column over the lifetime of the instrument. However, analysis of the L2 products retrieved from GOME-2A have suggested that the precision and quality of the DOAS fits have appreciably declined over the instrument's lifetime (Ditky and Richter, 2011). We have added this citation to the manuscript

*Wouldn't your statement "which suggests that coverage losses or instrument degradation are not significant influences on model accuracy or precision" here and in other places contradict your*

*discussions regarding SCIAMACHY (less sampling due to global coverage in 6 days) and GOME-2 (more cloudy pixels)?*

The statement in the manuscript refers to Models 1 and 2, which exclusively used OMI data, and so are not as influenced by coverage losses caused by cloudy pixels or temporal sampling. The perceived lack of influence on the model quality refers to these models only. We have altered the manuscript to better discuss this.

*I wonder how temperature can be a proxy for photochemical dissociation of NO2. Shouldn't it be actinic flux?*

We agree with this assessment, and have altered the manuscript to correct this.

*What is your measure for your model accuracy? Why are improvement in R2 and decrease in RMSE not considered for model improvement?*

Our main measure for model accuracy in this work was the CV gradient and bias, as this is a statistical model and so should better reflect the in-situ concentrations used to produce it. While the inclusion of ERA-Interim data does increase the model $R^2$ and decreases the RMSE, we determine that the overall accuracy of the model is not improved by the addition of this dataset.

*What is the logic behind applying daily average profiles instead of early-afternoon profiles that are more relevant for OMI? Could this be the reason for low correlation between OMI and in-situ observation?*

Daily average profiles were used to provide a comparative reference for the daily average $NO_2$ concentrations provided by the LUR models and the in-situ data. The analysis in this section was repeated using profiles modelled by MACC at 2:00 PM local time (the closest time available to the OMI overpass), with similar results. This suggests that the comparatively poor performance of MACC may be because of inaccurate emission data, rather than using a profile from a specific time of day.

*Deviation of red curve (fitted line) considerably from data points may suggest that the term in Eqn 4 that accounts for seasonal variation over time may not have been properly applied. Visually, the area under the curve passing through the points seems decreasing over time, consistent with the trend in emissions. Please check your calculation of trend. Please show the trend in OMI column data as well.*

We have repeated the analysis, and have obtained the same result. Please note that the damped oscillation term, ξ, may cause greater deviations from the fitted seasonal variation towards the end of the dataset. We found that fitting Eqn 4 to the OMI column data resulted in a statistically insignificant trend of: -2.52% $yr^{-1}$, and have added this result to the manuscript.

*How does the change in precision result in negative bias in surface concentration?*

It has previously been shown by Kim et al (2016) that the satellite footprint size can cause a smoothing of sub-pixel plumes over urban areas, and so the resulting retrieved column may be an underestimate of the true value. We have amended the manuscript to include this citation and better wording.

[revised manuscript text omitted]

---

## Author Comment (AC2) · 13 May 2017

We thank the reviewer for their valuable comments. We have amended the manuscript in light of their suggestions, and will address the most pressing of these herein.

*I'm not convinced that the authors have developed the land use regression approach sufficiently for it to have novel general implications. The authors acknowledge their model is similar to a previously developed mixed-effects model, and thus the advancement here is largely in its application to a different region.*

We disagree with this assessment for a number of reasons:

- To our knowledge, a purely statistical approach to modelling urban $NO_2$ with satellite data at such small spatiotemporal scales has previously not been attempted. The good agreement with in-situ data suggests that our approach is capable of estimating urban concentrations at small spatial scales. Because of this, we believe that our approach offers a unique perspective on urban air quality which can be readily compared to existing CTM-based approaches for validation purposes, particularly with the prospect of higher spatial resolution data becoming available from Sentinel-5P.
- While the mixed-effects approach was indeed trialled in a previous publication, the fixed parameters used in this work were determined using a much more robust selection procedure in order to properly quantify the effect these variables have on the model fit.
- We have also for the first time shown the limitations of this approach, particularly when trying to account for the diurnal cycle through using multiple satellite instruments.
- Despite the small number of in-situ monitoring stations available, we found that the mixed effects approach allows for much better forecasting of daily $NO_2$ concentrations when satellite data was used, as shown in Tables 3 and 4.
- Finally, the trend estimation and comparison with the bottom-up emission inventory suggests that existing pollution control measures enacted by the HKEPD alone are insufficient for improving the air quality of Hong Kong, potentially because of transported pollution from mainland China. This is in line with the conclusions drawn by Xue et al (2014), and more recently in Wang et al (2017).

*I am not yet satisfied that 11 monitoring stations are sufficient to build a national-scale LUR model. As they admit, the coverage is predominantly urban. We have no sense of what "dynamic range" exists in the predictor variables across these urban sites, and whether there is enough to build a robust model. My problem is that while there are many individual daily observations, parameters like road length, urban area coverage, population density, vegetation area coverage, and elevation will not vary day-to-day. Thus, they only have 11 unique data points for each variable. This strikes me as an extremely small sample.*

We agree with the reviewer that the number of in-situ stations provided in this work is smaller than the number typically used in LUR models of urban areas (Hoek et al, 2008). At the time of this work no other surface concentration data was available from the Hong Kong SAR outside of the 11 stations quoted in this work. However, it is not entirely without precedent; Li et al (2010) used a similar number of monitoring stations in their LUR model of Jinan, China. Because of this, we contend that 11 stations are sufficient to base a local model for a region as small as Hong Kong.

Additionally, the spatial patterns predicted by our model are visually similar to the $NO_2$ concentrations predicted by the LUR model created by Lee et al (2017), who used 95 measurement locations to derive their model – we have added this citation to the manuscript. As shown in Tables 4 and 5 the mixed-effect models show very good temporal agreement with the in-situ data compared with the reference

model, suggesting that the use of satellite data at least partially compensates for the sparse in-situ station placement.

The main source of spatiotemporal information in the model is the $S/\Omega$ relationship derived from the in-situ and satellite measurements, while the other parameters are used to only give a local context for possible $NO_x$ emission sources and sinks. Accurate accounting of the temporal evolution of such sources and sinks would require high-resolution information such as traffic volume and industrial activity to be added to the model, which was unavailable at the time of this work. While it is possible to derive temporal variation in emission sources from satellite data (e.g. updating existing emission inventories through model assimilation, e.g. Mijling et al, 2012), it is unlikely that existing CTMs or satellite instruments can provide the spatial resolution necessary to meaningfully enhance the models developed in this work.

*Another test would be to try building the model(s) by holding back specific groundstation locations (and not just a percentage of the available data from each location, since this does not test the sensitivity to the loss of a ground station location). Was this the purpose of Section 3.4? I was a bit confused about what this section was presenting. In general, I am not satisfied that a cross-validation approach involving removing 20% of all the data is a good enough test. It strikes me that a better test would be to remove entire stations instead. Keeping any observations from every station basically means that most of the predictor variables aren't actually getting tested (since these are not changing day-to-day)*

As stated previously, the key parameter of the spatiotemporal variation predicted by the LUR model is the $S/\Omega$ relationship derived from the satellite and in-situ data. Because of the limited number of in-situ stations available, removing entire stations from the dataset for validation will remove essential spatial information from the model, which will significantly bias the validation result depending on the station placement. Therefore, as previously shown in other LUR-related studies (Johnson et al (2010); Wang et al, (2016)), the "leave-one-out-cross-validation" approach suggested by the reviewer overestimates the LUR model performance compared to the "k-fold-cross-validation" approach used in this work.

*Finally, I think the introduction to Section 2.3 about mixed effects LUR could use more detail. What would the formulation of a "fixed"/"random" effects model look like in comparison? What is the advance in the mixed effects model? As someone who is not overly familiar with the LUR approach, it is not really clear to me what the mixed effects model actually captures and how it is calculated (despite the inclusion and explanation of Equation 1). The authors should assume that some of the ACP audience will not be familiar with LUR.*

We thank the reviewer for this advice, and have expanded the explanation in Section 2.3.

*Can you describe in a more detail the network of DOAS vs. in-situ measurements? How many of each? Do they have the same length of coverage? Are the data for both available online? Can you show which ones are which on the map? Are the chemiluminescence measurements via catalyst or LED conversion?*

At the time of this reply we have been unable to determine from available literature and metadata which stations contain the aforementioned instruments, nor have we been able to find out their mode of operation.

[revised manuscript text omitted]